# Ultra-sensitive monitoring of leukemia patients using superRCA mutation detection assays

Lei Chen [1,3✉], Anna Eriksson[2], Simone Weström[1], Tatjana Pandzic[1], Sören Lehmann[2], Lucia Cavelier[1,4] & Ulf Landegren [1,4✉]

Rare tumor-specific mutations in patient samples serve as excellent markers to monitor the course of malignant disease and responses to therapy in clinical routine, and improved assay techniques are needed for broad adoption. We describe herein a highly sensitive and selective molecule amplification technology - superRCA assays - for rapid and highly specific detection of DNA sequence variants present at very low frequencies in DNA samples. Using a standard flow cytometer we demonstrate precise, ultra-sensitive detection of single-nucleotide mutant sequences from malignant cells against up to a 100,000-fold excess of DNA from normal cells in either bone marrow or peripheral blood, to follow the course of patients treated for acute myeloid leukemia (AML). We also demonstrate that sequence variants located in a high-GC region may be sensitively detected, and we illustrate the potential of the technology for early detection of disease recurrence as a basis for prompt change of therapy.

[1] Department of Immunology, Genetics and Pathology, Science for Life Laboratory, Uppsala University, SE-752 37 Uppsala, Sweden. [2] Department of Medical Sciences, Uppsala University, SE-751 05 Uppsala, Sweden. [3] Present address: Rarity Bioscience AB, SE-752 37 Uppsala, Sweden. [4] These authors jointly supervised this work: Luica Cavelier, Ulf Landegren. ✉email: lei.chen@raritybioscience.com; ulf.landegren@igp.uu.se

Tumor-specific mutant DNA can serve as a highly specific class of biomarkers for monitoring malignant disease and responses to therapy[1]. Increasingly, frequently mutated genes or total genomes are being sequenced in tumors as a matter of routine, furnishing lists of tumor-specific somatic mutations that characterize individual tumors[1–4]. Analysis of such sequence variants places stringent demands on the techniques used. Given sufficiently sensitive, specific and inexpensive techniques to detect rare point mutations, it will be possible to monitor the course of disease and responses to therapy for practically any tumor patient. Suitable material to investigate for trace amounts of somatic mutations specific for the patients' malignant clones include cell-free DNA in body fluids and cells from blood or bone marrow (BM).

While loss of function mutations in recessive oncogenes can be unique to individual patients, other mutations recur in many patients. Such mutations often increase the activity of their gene products and stimulate tumor growth, and they may prove actionable by signaling responsiveness to targeted drug therapies[5]. Because of their prevalence among patients, even a limited repertoire of assays for specific recurrent mutations could serve to monitor the course of malignant disease in many patients.

A case in point is acute myeloid leukemia (AML), a severe hematological condition with poor prognosis and characterized by clonal expansion of early myeloid blasts. Although intensive combination chemotherapy induces remissions in most patients, the majority of those patients will ultimately relapse and succumb to the disease within two years[6,7]. Genetic abnormalities constitute powerful prognostic factors in AML and several recurrent mutations, including mutations in *NPM1*, *FLT3* and *TP53*, are used in disease risk stratification. Furthermore, insight into the molecular basis of AML have enabled several targeted therapies to emerge in recent years directed against these recurrent genetic aberrations[7].

A number of the most common mutations in AML occur in genes involved in epigenetic gene regulation, and several of them have been reported as early key events during leukemogenesis[8], including ones affecting *DNMT3A*, isocitrate dehydrogenase 1 and 2 (*IDH1* and *2*), and chromatin modifiers such as *ASXL1*. Mutations in these genes that lead to epigenetic changes are also frequently seen in other hematological disorders such as myelo-dysplastic syndromes (MDS), as well as in solid tumors, including colorectal cancer and brain tumors[9–11].

*IDHs* are important regulators of the normal citrate metabolism and *IDH1/2* genes are mutated in 15–33% of AML cases, causing a shift from normal a-ketoglutarate production to generation of the oncometabolite 2-hydroxyglutarate[8]. Two targeted therapies (ivosidenib and enasidenib) have recently been approved by the FDA for treatment of AML with *IDH1*- or *IDH2*-mutations, respectively.

Relapse of disease upon treatment constitutes one of the major challenges in AML today and improved understanding of its causes as well as new techniques to predict risk of relapse or detect relapse at the earliest time possible are both highly warranted. Multiple studies have consistently shown the prognostic importance of assessing measurable residual disease (MRD) for estimating remaining leukemic cells after therapy, with MRD-negativity highly prognostic for beneficial disease outcome[12]. The most widely used techniques are multiparameter flow cytometry to identify antibody-stained malignant cells and real-time quantitative polymerase chain reaction (qRT-PCR)[12,13] for fusion transcripts, applied on BM samples. Results from such MRD analyses can guide treatment decisions in AML patients, particularly the decision whether or not a patient should undergo an allogeneic stem cell transplantation[7]. Although chemotherapy has been the backbone standard of care in AML for the last 40 years, we are now moving towards a more targeted therapeutic landscape with the goal of tailored precision treatment for each patient. Accurate detection of MRD will have an increasing value to guide therapeutic decisions as more targeted treatment options become available to choose from. It would be of significant practical advantage, not least for the patients, if biomarkers for MRD could be detected with ultra-high sensitivity in readily available peripheral blood samples rather than in bone marrow.

Several methods are in use to detect sequence variants present at low frequencies in DNA samples[14–16]. For example, variants of the basic PCR procedure can provide extreme selectivity for single nucleotide variants, such that also very small numbers of mutant DNA sequences can be detected against a considerable excess of the normal sequence, but these methods fail to provide quantitative results[17–19]. Rare mutant molecules can be accurately enumerated by compartmentalizing many single-molecule amplification reactions for droplet digital PCR (ddPCR)[16,20–22]. Deep DNA sequencing of DNA molecules that have been tagged at either one or both strands with unique molecular identifiers (UMIs) to exclude artificially introduced sequence changes, has been used to estimate numbers of mutant sequences in patient samples at abundance levels as low as one in 100,000 by devoting millions of reads per mutation site[14,23–25]. See Supplementary Table S1 for a comparison among some of these methods.

Here we demonstrate an approach that enables ultrasensitive, precise quantitation of mutant DNA sequences in a convenient format—superRCA—and we use this to monitor patients with hematologic malignancies. DNA sequences of interest, known to be mutated in a patient's malignant cells, are first enriched by targeted PCR amplification from a patient sample (Fig. 1). The amplified sequences are next converted to DNA circles that are subjected to rolling-circle amplification (RCA). Padlock probes specific for mutant or wild-type sequences are then used to probe the repeated sequences of the RCA products with exquisite specificity, followed by RCA of the circularized probes. The large DNA clusters that result from each starting DNA circle are referred to as superRCA products. They can be visualized using

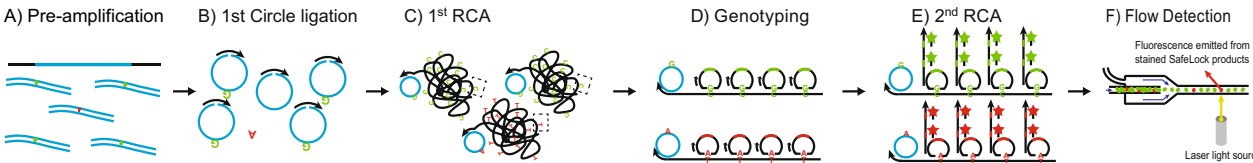

**Fig. 1 Generation of superRCA amplification products. A** DNA sequences of interest in a sample are amplified by PCR. **B** Amplified strands are converted to single-stranded DNA circles via templated ligation of their 5′ and 3′ ends. **C** Oligonucleotides that template the circularization reactions next serve as primers for RCA reactions. **D** The RCA products are then interrogated with padlock probes specific for mutant or wildtype sequences. **E** Ligated padlock probes, wound around the RCA products, thereafter template secondary RCA reactions, primed by an added oligonucleotide. **F** For each starting DNA circle the reaction gives rise to large clusters of mainly single-stranded DNA objects, called superRCA products. Up to a million fluorescence-labeled hybridization probes can bind each of the mutant- or wildtype-specific products, allowing efficient counting via e.g., standard flow cytometry.

fluorophore-labeled hybridization probes and counted as individual, brightly fluorescent mutant- or wildtype-specific objects using a standard flow cytometer or by microscopy. The majority vote-based genotyping of hundreds of concatenated copies of the target sequences in individual RCA products is the key feature explaining the high fidelity of genotyping, despite a generally 0.1–0.5%[26] error rate per target sequence copy in ligase-mediated genotyping approaches[27,28].

We demonstrate the suitability of the superRCA approach for digital detection of even very rare mutant DNA sequences, including deletions, insertions and single nucleotide exchanges, also in challenging sequence contexts e.g in high GC% segments, by investigating samples from patients with blood malignancies. We illustrate that the great sensitivity for mutant sequences has the potential to enable detection of MRD at very low levels allowing for the early detection of recurrent disease.

## Results

### Individual single-stranded DNA circles yield large DNA clusters - superRCA products - via two consecutive RCA reactions.
To produce superRCA products, DNA sequences of interest are first enriched by PCR, and amplified strands are ligated to form single-stranded DNA circles (Fig. 1, see also Fig. S1). Long concatemeric products are generated from the starting DNA circles by RCA. Next, added pairs of padlock probes, complementary to the hundreds of copies of either mutant or wildtype sequences in each of the primary RCA products, are ligated in a target-specific manner. Upon successful target recognition the padlock probes form circular DNA strands, wound around the RCA products, and they are then subjected to a second RCA reaction.

Probing of the primary RCA products, each containing several hundred copies of either the normal or mutant target sequence, yields an extreme degree of sequence selectivity. This is so because the occasional target misidentification by an allele-specific padlock probe remains undetectable as long as the vast majority of the copies in any primary RCA product are accurately genotyped.

After the initial PCR, the above procedure is operated by a series of additions to microtiter wells and incubations. For each starting DNA circle this produces a large DNA cluster, termed superRCA product. In the experiments shown herein both the starting DNA circles and the padlock probes used for genotyping are of the order 100 nt in length. Given the rate of polymerization by the Phi29 polymerase 1 h RCA reactions yield RCA products composed of several hundred to a thousand complements of each starting DNA circle. Each easily detectable, clonal superRCA product, generated via the two consecutive RCA reactions, therefore contains up to $1000 \times 1000$ monomer sequences or around one hundred megabases of DNA, and they each have molecular weights that can reach tens of GigaDaltons.

Figure 2A illustrates a first-generation RCA product and a superRCA product that has been subjected to two generations of RCA, visualized by scanning electron microscopy. The image illustrates that superRCA products reach dimensions of several micrometers of the order of human cells. Figure 2B demonstrates results of an experiment where the replication of different concentrations of first- and second-generation RCA products was measured in real time by monitoring the fluorescence of molecular beacons[29] added to the amplification reactions. The experiments showed that a full thousand-fold amplification can indeed be achieved in both the first and second 1 h RCA reaction, resulting in a total million copies of a tag sequence per superRCA product.

We used fluorescence microscopy to investigate whether individual superRCA products do in fact derive from single starting DNA circles without the need for compartmentalization of the reactions. superRCA products, generated from two kinds of starting DNA circles were stained in two distinct colors revealing no mixed-color products, thus confirming that each superRCA product derives from a single starting DNA circle (Fig. 2C). The prominent, brightly fluorescent reaction products allowed counting and distinction of individual amplification products of single molecules across wide fields of view at low magnification (20X). For the pure wild-type or mutant samples only green and red signals, respectively, were observed demonstrating the unambiguous identification of wild type or mutant samples. A few yellow products observed for the 10:1 condition in Fig. 2C are due to the random colocalization of the high concentration of wild-type and mutant amplification products deposited on the glass surface.

**superRCA products can be enumerated via flow cytometry**. We found that individual superRCA reaction products were sufficiently large and bright to be recorded by standard flow cytometry, normally used for analyzing fluorescence-labeled cells. The approach allows superRCA products, each of which represents a single starting DNA circle, to be conveniently counted for any molecular assay that results in the formation of DNA circles, and many such assays exist[30–33]. In this manner, millions of products can be digitally scored in a matter of minutes using generally available instrumentation, thus offering excellent quantitative precision over wide dynamic ranges (Fig. 3A, Supplementary Figs. 2 and 3). We also illustrate that the repeated sequences of the secondary RCA products allow for flow cytometric analysis of superRCA products labeled with distinct combinations of two fluorophores (Fig. 3C).

**Detection of ultralow frequency point mutations in AML**. IDH mutations, frequently seen in AML patients, along with 8 other AML-related mutations, were targeted by a superRCA 12-plex panel including 9 single-nucleotide mutations (IDH1 p.R132C, IDH1 p.R132H, IDH2 p.R140Q, IDH2 p.R172K, TP53 p.R248Q, PTPN11 p.A72T, DNMT3A p.S714C, DNMT3A p.R882C and BCORL1 p.Q1039*), 2 insertions (NPM1 p.W288fs and ASXL1 p.G646fs*12) and 1 deletions (BCOR p.M1641fs*50) to monitor MRD in AML patients. Genes commonly mutated in AML and MDS, mutations of different types, aberrations with prognostic impact and mutations where targeted therapy are available or underway were included in the analysis. 10 co-amplified PCR amplicons were designed, allowing parallel detection of 12 mutations present in 9 genes from the same DNA sample. The reaction products were then split and each of the 12 mutations were interrogated in individual superRCA reactions.

The IDH mutations were used to benchmark the analytic performance of our method. We first prepared 4-fold serial dilutions of genomic DNA from cell lines carrying the respective mutations into wild-type genomic DNA lacking the mutation. The total range of mutant allele frequencies (MAFs) in the dilution series spanned from $10^{-2}$ to $10^{-5}$, with pure wild-type genomic DNA sample in the leftmost data points representing negative controls. Each DNA dilution was analyzed either by the superRCA procedure or using a commercial droplet digital PCR (ddPCR) assay as recommended by the manufacturer (Fig. 4). To optimize detection sensitivity, 330 ng genomic DNA, equivalent to 50,000 diploid cells, was used in the superRCA assay. In initial ddPCR, the use of 330 ng genomic DNA input (pooled from 10 ddPCR reactions with 33 ng gDNA per ddPCR reaction) resulted in similar levels of false positive events as 100 ng genomic DNA input in the IDH assays (Supplementary Figs. 4 and 5). We therefore elected to use 100 ng input for the ddPCR assay throughout the study. The superRCA assays successfully detected mutant DNA in up to a 100,000-fold excess of normal DNA.

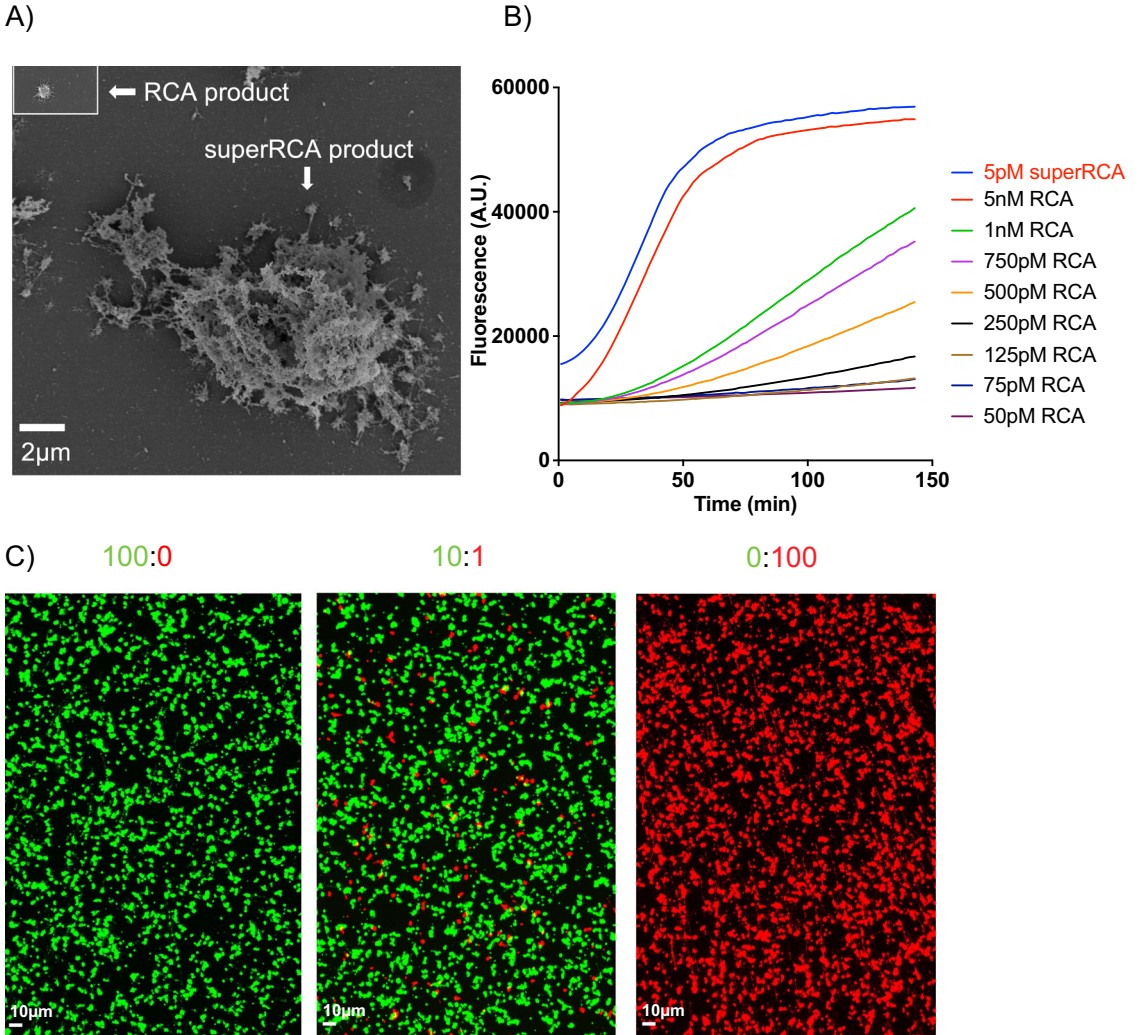

**Fig. 2 Properties of superRCA reaction products. A** Scanning electron micrographs illustrating the relative sizes of a first-generation RCA product (inset) and a superRCA product having undergone two generations of RCA. **B** Real-time monitoring of first- and second-generation RCA reactions illustrates the extent of amplification in each generation. RCA replication of 5 pM superRCA products, accumulate fluorescence at the same rate as first generation RCA products templated by 5 nM circularized padlock probes. The experiment demonstrates that the generation of monomer repeats during superRCA proceeds at a rate approximately 1000-fold faster than in a first RCA, as evidence that each first-generation DNA circle gives rise to around 1000 second-generation DNA circles. **C** Individual superRCA products, originating from single starting DNA circles as visualized by fluorescence staining of products of three different reactions, starting from two DNA circles yielding green or red superRCA products in proportions of 1:0. 10:1, or 0:1. No mixed color products are observed, identifying these as digital objects.

The ddPCR data revealed substantially greater variability and insufficient sensitivity for very low-frequency mutations. Representative raw superRCA and ddPCR data for samples with the mutation IDH2 p.R172K spiked at low MAF are shown in Figure S6. We also analyzed four diagnostic samples from patients UPN 113, UPN 125, UPN 130 and UPN 131 with the 12-plex AML panel to demonstrate the feasibility of detecting multiple mutations within each single patient sample (Supplementary Fig. 8).

**Analysis of a set of 44 patients with myeloid malignancies for the presence of four IDH mutations**. We proceeded to analyze the four IDH mutations shown in Fig. 5 in BM aspirates from a cohort of 22 patients with myeloid malignancies, including AML, MDS and myeloproliferative neoplasia. All patients were investigated by superRCA, ddPCR and by next-generation sequencing (NGS). The samples were collected at diagnosis when all patients were acutely ill with high proportions of malignant cells in their

BM. IDH gene mutations were recorded for all patients using targeted NGS assays as part of the clinical workup. We found that all patients carried at least one of the investigated mutations, while two patients also carried a second mutations in IDH at a lower frequency. The MAF values of these diagnostic samples as recorded using superRCA were in good agreement with the results of both NGS and ddPCR analysis (Supplementary Fig. 7A–F).

Since the superRCA assay for the IDH mutations was configured to evaluate all four mutations from the same PCR reaction, a total of 330 ng DNA was used per patient to allow for sensitivities down to 1 in 100,000 haploid genomes, while the ddPCR assays required a total of 400 ng DNA to investigate the same four targets in four separate reactions with lower sensitivity. The absolute limit of detection (LoD) of superRCA and ddPCR were determined by spike-in experiments (Fig. 4) The Initial PCR cycles served to expand the chance to detect the mutant DNA molecules in the sample for analysis of several mutations in parallel as herein or in multiplex reactions. For the one in 100,000

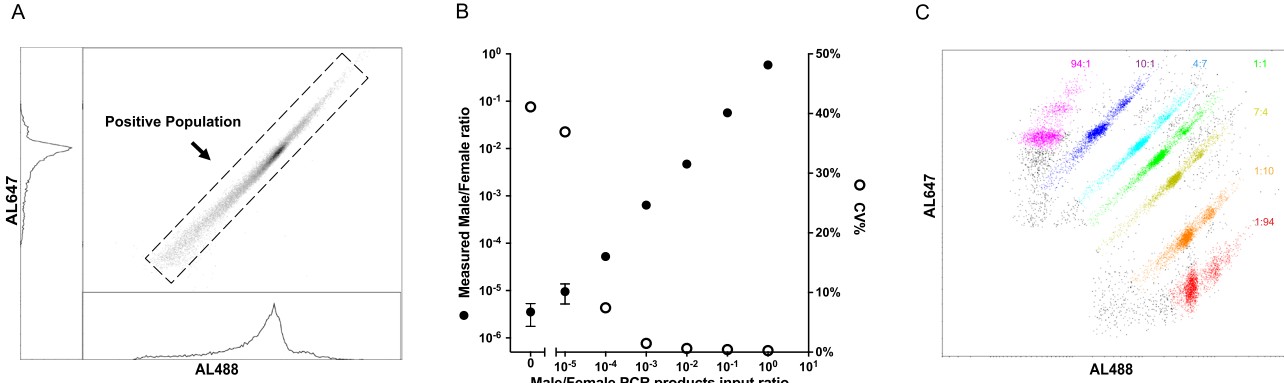

**Fig. 3 Analysis of superRCA products via flow cytometry. A** Flow cytometric analysis of superRCA products visualized using two hybridization probes, one labeled with the fluorophore AL488 and one with AL647. **B** Flow cytometric analysis of superRCA products resulting from detection of a target sequence from the Y-chromosomal SRY gene compared to superRCA products for a sequence from the autosomal β-globin gene. Filled circles (·) identify the recorded proportions of superRCA products for the indicated input ratios of the two gene sequences. Open circles (o) denote the coefficients of variation (CVs) of triplicate measurements. Y chromosome-specific superRCA products were still detected above background at a dilution of one in 100,000. $n = 3$ biological replicates were analyzed for this data set and the data are presented as mean value ± SD. **C** The large numbers of repeated sequences per superRCA product allows individual objects to be stained with combinations of two fluorescence-labeled hybridization probes, as illustrated by plotting in false colors results for superRCA products stained using the indicated proportions of two oligonucleotides labeled with AL488 or AL647 and then pooled for flow cytometry analysis.

MAF (mutant allele frequency) data point in the spike-in data experiments (we used a total DNA input of 660 ng, which corresponds 2 mutant molecules in 200,000 haploid genomes).

For a second cohort consisting of 24 AML patients, a total of 33 ng DNA were used to detect the IDH mutations. Also for this group of patients the ddPCR and superRCA assays confirmed the presence of the same IDH mutations as seen by NGS and at comparable frequencies. The ddPCR signals for the IDH1 p.R132H mutation for patients 204 and 221 were near the limits of detection (0.14% and 0.12%, respectively) and are probably incorrect as they were not observed by superRCA (Supplementary Fig. 9).

**Monitoring AML patients over time using superRCA, NGS and ddPCR assays.** Three AML patients for whom consecutive samples were available were analyzed at several time points using the three methods to explore the suitability of superRCA probing for monitoring the course of disease (Fig. 6). The first patient (UPN124) had presented with low blood counts three and a half years prior to AML diagnosis, but BM analysis at that time failed to confirm any hematological diagnosis. The IDH2 p.R172K mutation was identified in a BM sample using a targeted NGS detection assay at AML diagnosis, and intensive chemotherapy was initiated. The leukemic clone remained unchanged after initial chemotherapy, but a complete remission was achieved after switching to a combination of azacitidine and venetoclax (complete remission, i.e., <5% blast cells morphologically) with MRD measured by analysis of antibody-stained cells using flow cytometry of <0.1% (CD34 +, CD117 +, CD33 +, CD13 heterogenous HLA-DR + and CD56 +). Allogeneic stem cell transplantation (SCT) was performed on day 126, and the patient has remained symptoms-free for a follow-up period of 21 months post SCT. All three assays reported similar proportions of mutant copies of the IDH gene in a BM sample taken at diagnosis (Fig. 6A). The IDH mutation was still presents at a considerable frequency in a peripheral blood sample taken 6 weeks prior to the SCT as recorded via ddPCR ($1.66 \times 10^{-3}$) and superRCA ($1.27 \times 10^{-3}$) even though the patient was clinically MRD-negative as determined by flow cytometry of antibody-stained bone marrow cells (MRD <0.1%). The patient was monitored after the SCT by repeated analyses of blood and BM samples through ddPCR and

superRCA, and at some timepoints also NGS. All the six collected samples revealed undetectable levels of IDH2 p.R172K mutations using all three assays, indicating that the patient has remained in remission. We also located the first BM sample from this patient, collected three and half years before the diagnosis of AML was made when the patient first presented with slightly lowered blood counts. Both superRCA and ddPCR assays recorded even higher levels of the IDH2 p.R172K mutation in this sample compared to the sample taken at AML diagnosis, confirming the presence of clonal hematopoiesis in the setting of peripheral cytopenia, also named clonal cytopenia of undetermined significance, prior to the AML diagnosis for this patient.

For another patient (UPN125), diagnosed with AML with malignant cells showing morphological signs of a myelodysplastic character, three different mutations were followed during the retrospective study with single portion DNA samples. The NGS analysis of a BM sample at diagnosis revealed a IDH2 p.R172K mutation at a MAF of 0.39, DNMT3A p.R882C at a MAF of 0.38, and BCORL1 p.Q1039* at a MAF of 0.14. After treatment with 3 courses of intensive chemotherapy, the IDH2 p.R172K mutation was undetectable by NGS on day 97, and the patient was characterized as being in clinical remission. By contrast, retrospective analyses of the same sample by superRCA and ddPCR assays recorded the IDH2 p.R172K mutation at frequencies of $1.79 \times 10^{-3}$ and $2.39 \times 10^{-3}$, respectively, demonstrating early detection of the continued presence of the disease. (Fig. 6B). All subsequent samples, taken from day 182 and onwards were positive by all three assays with MAF values between 0.01 and 0.1, even though the patients peripheral blood counts only started to decrease around day 230. The patient has subsequently relapsed but in a MDS phase without blast excess. Treatment was restarted with azacitidine on day 565 after initial diagnosis. The BCORL1 p.Q1039* mutation was initially detected by superRCA at a MAF of 0.12 in the diagnostic sample at Day 0 and then decreased to background levels of NGS and below $10^{-4}$ in the superRCA assay in a remission BM sample collected at Day 97 after 3 courses of intensive treatment. The mutation has remained at similar levels in subsequently collected samples (Fig. 6C). The DNMT3A p.R882C mutation was initially detected by superRCA at a MAF of 0.44 in the day 0 diagnostic sample. The MAF remained at significant levels by NGS (MAF $2.1 \times 10^{-2}$) and

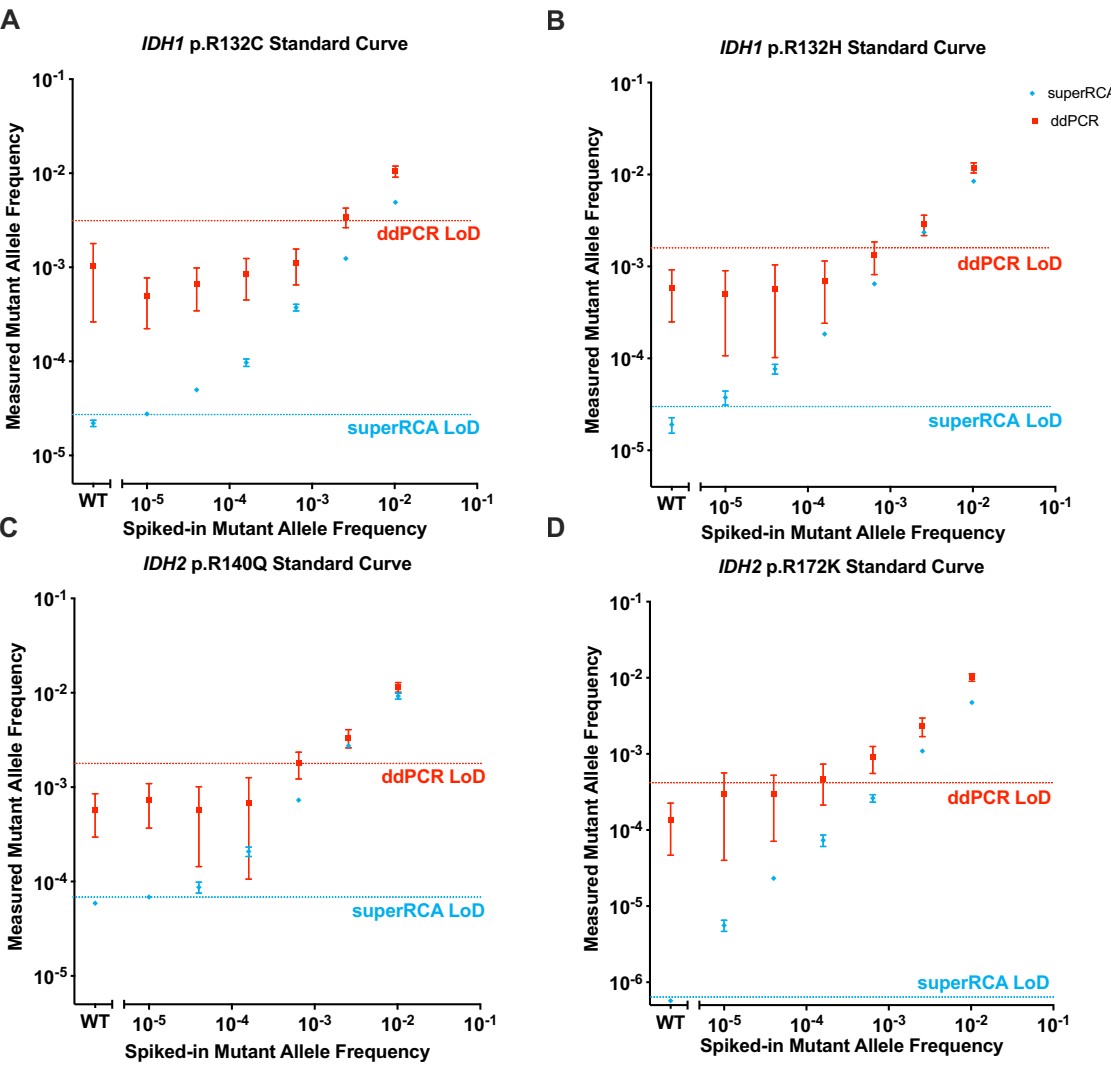

**Fig. 4 Comparison between analytical performance of superRCA and ddPCR assays for four IDH mutations.** Analysis of four-fold serial dilutions of genomic DNA from cells with **A** *IDH1* p.R132C, **B** *IDH1* p.R132H (**C**) *IDH2* p.R140Q, and **D** *IDH2* p.R172K mutations using the superRCA procedure or ddPCR. Genomic DNA samples with mutant genomic DNA were serially diluted in wildtype genomic DNA and divided into two portions each for and analysis by superRCA or ddPCR. 330 ng DNA, corresponding to 100,000 human haploid genomes (or a little less than 50,000 cells, considering that the cells are cycling and may have partly replicated their genomes), were analyzed per sample for the superRCA assay, while a total of 100 ng DNA divided in two replicate reactions, corresponding to 15,100 human haploid genomes (corresponding to 7,500 diploid cells) were used for ddPCR, according to the manufacturer's instructions. $n = 3$ biological replicates were analyzed for superRCA and $n = 2$ biological replicates were analyzed for ddPCR assay set. The data are presented as mean value ± SD. The limit of detection (LoD) for ddPCR and superRCA assay for each analyte was calculated as *LoD = Mean(wildtype) + 3 × SD(wildtype)* and presented in the figures with horizontal dashed lines.

superRCA (MAF $2.1 \times 10^{-2}$) in a BM sample collected during clinical remission (Day 97) after 3 courses of intensive treatment, consistent with a preleukemic clone persistent in morphological complete remission[13]. This mutation continued to be present at high levels during further retrospective monitoring (Fig. 6D).

A third patient (UPN126), previously diagnosed with primary myelofibrosis associated with a JAK2-mutation, transformed to a secondary AML 2 years after the initial diagnosis (Fig. 6E). NGS at AML diagnosis revealed the presence of an *IDH2* p.R140Q mutation alongside the previously known JAK2-mutation. Retrospective analysis of the initial bone marrow sample in the myelofibrosis stage was negative for the *IDH2* p.R140Q mutation when analyzed by both superRCA and ddPCR assays, indicating a clonal evolution upon AML transformation. The patient achieved a complete remission after the first course of intensive chemotherapy but remained MRD-positive at approximately 2% (measured by flow cytometry of bone marrow cells with CD34,

CD117, CD13 and DR+ markers) after 3 courses of intensive chemotherapy. An allogeneic SCT was performed on day 139. The patient was subsequentially monitored by analyzing consecutive blood and BM samples through ddPCR and NGS and all six samples collected post-SCT revealed undetectable levels of the *IDH2* p.R140Q mutation by both assays, indicating that the patient has remained in remission with a follow up of more than two-year post SCT. The blood sample collected 3 months post SCT was positive by superRCA but under the detection limit for ddPCR. In routine care, the finding of a positive result for the *IDH2* p.R140Q mutation is important, as it would prompt the discontinuation of immunosuppressants to boost the immunological effect of the transplant.

**Analysis of a mutation located in a GC-rich region in AML and MDS.** Myelodysplastic syndromes (MDS) comprise a group of

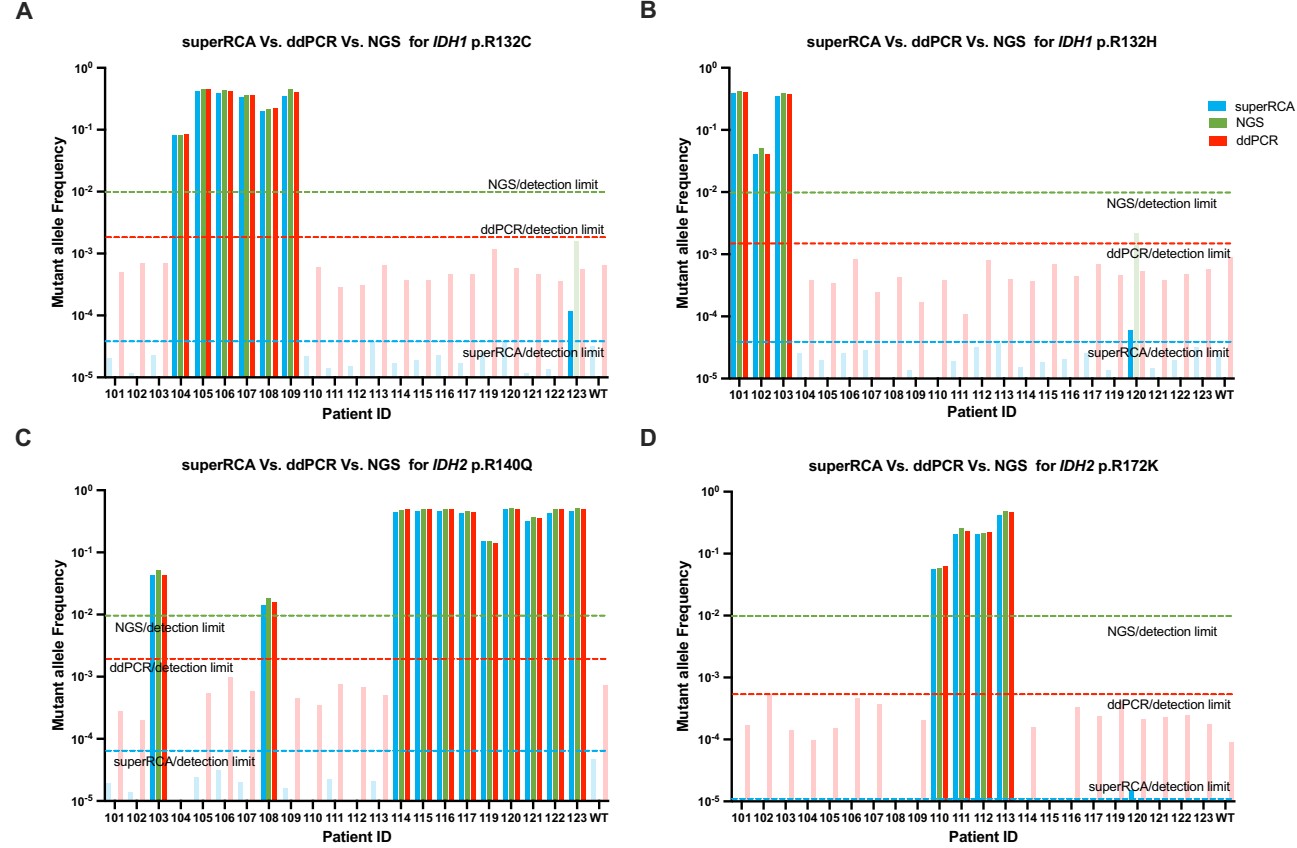

**Fig. 5 Benchmarking of superRCA, NGS and ddPCR assays in BM aspirates from AML patients.** Samples from 22 patients, all with high proportions of malignant cells in BM, were selected for the benchmark study. 330 ng genomic DNA was used per patient for the multiplexed superRCA assays. After the pre-amplification step in the superRCA protocol, each sample was subjected to parallel analysis of **A** *IDH1* p.132 C, **B** *IDH1* p.R132H (**C**) *IDH2* p.R140Q and **D** *IDH2* p.R172K. For the ddPCR assay, four separate IDH mutation assays were investigated per patient, using 100 ng gDNA divided into two replicates for each mutation. The NGS data were analyzed at diagnosis of the patients, revealing the IDH mutation status for each patient sample included in this benchmark study. Values recorded below the detection thresholds for the ddPCR and superRCA assays are shown in light colored bars. $n = 1$ biological replicate was analyzed for superRCA, ddPCR and NGS for this panel, and the data were presented as single measurement. Samples for all three analysis methods were collected at diagnosis of the patients.

disorders where the BM produces insufficient numbers of healthy blood cells. Roughly 30% of MDS patients progress to AML. The *ASXL1* p.G646fs*12 mutation is considered a marker of poor prognosis in both MDS and AML[7,34]. This mutation is characterized by an extra G nucleotide inserted in a sequence of 8G nucleotides (GGAGGGGGGGG[-/G]TGGCCCGGGTG). The GC-content of the 20 bp surrounding the insertion site is 87%, precluding analysis by a commercial ddPCR assays. The detection threshold of an in-house NGS assay only reached a minimal MAF of 5%, and similar results have been reported in the literature[35]. To investigate the feasibility of using superRCA assays for analyzing this mutation at low frequencies, we prepared a fourfold dilution series of genomic DNA containing this mutation in genomic DNA with the wild type sequence (Fig. 7A). The superRCA assay successfully revealed the GC-rich mutation even at the lowest investigated concentration of $2 \times 10^{-4}$ in 33 ng aliquots of genomic DNA.

We applied the superRCA assay for the GC-rich *ASXL1* p.G646fs*12 mutation, previously detected by NGS in samples collected from four patients with myeloid malignancies (AML or MDS) (Fig. 7B). BM samples from the two patients UPN113 and UPN127 were positive for the mutation at frequencies of 0.32 and 0.24, respectively, corresponding to the NGS-results from the diagnostic workup. For the two remaining patients (UPN128 and UPN129), diagnosed with AML and acute promyelocytic

leukemia, respectively, NGS detected the *ASXL* p.G646fs*12 mutation at a MAF around the detection limit of 0.05, but the mutation was not observed when analyzed by superRCA, suggesting that the sequencing results may have been in error. The high background of NGS for the *ASXL1* p.G646fs*12 mutation has been previously reported[35,36].

The patient UPN113 was diagnosed with MDS with blast excess and treated with 4 cycles of azacytidine. The remaining high blast percentage of around 20% motivated a therapy shift to intensive chemotherapy, followed by an allogeneic SCT performed on day 253 after the initial diagnosis. superRCA analysis at one and three months post SCT revealed undetectable levels of the *ASXL1* p.G646fs*12 mutation in agreement with the patient's clinical remission. Nonetheless, this patient regrettably died of treatment-related causes on day 352, a stark reminder of the desirability of data about patient responses to optimally balance the therapy.

## Discussion

The need to demonstrate tumor-specific nucleic acid sequence variants against a great excess of closely similar sequences places stringent demands on the detection techniques used. The rare molecules must be detected with high efficiency, precisely because they are rare, and therefore must not be overlooked in limited

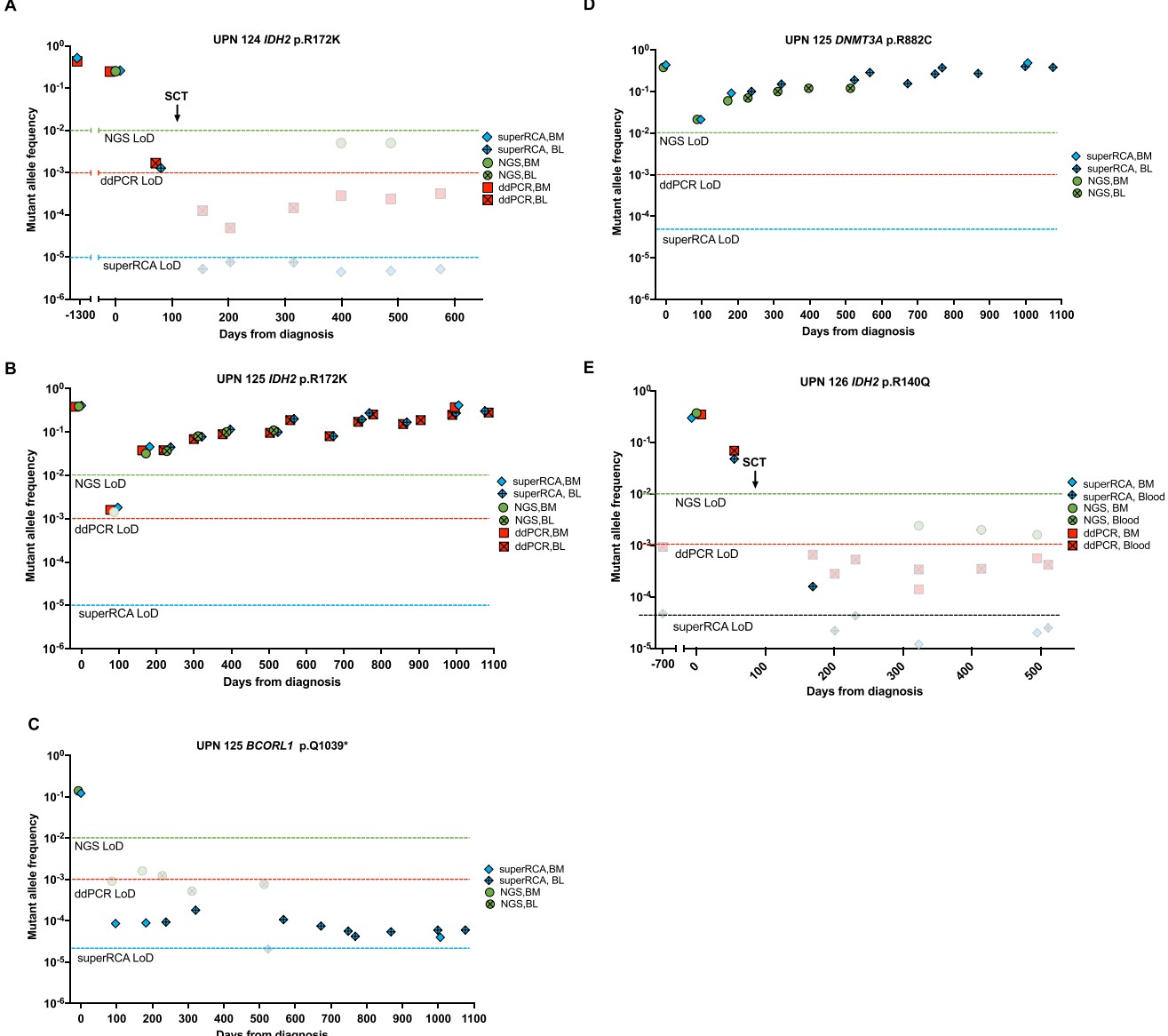

**Fig. 6 Follow-up of three AML patients using ddPCR, NGS and superRCA assay. A** Patient UPN124 was diagnosed with AML on day 0 with high levels of the *IDH2* p.R172K mutation, and drug therapy was initiated. SCT was performed on day 126, at which point considerable levels of the mutation remained, as recorded via superRCA and ddPCR, but all subsequent samples post-SCT have been negative, indicating a complete remission. A bone marrow sample taken three and a half years prior to AML diagnosis demonstrated high levels of the *IDH2* p.R172K mutation when retrospectively analyzed by superRCA and ddPCR assays. **B** Patient UPN125 was followed-up with respect to the *IDH2* p.R172K mutation. For UPN125, NGS of a BM sample taken at AML diagnosis (day 0) revealed high levels of *IDH2* p.R172K,mutation, and subsequent superRCA and ddPCR analyses of the same sample recorded similar levels of the mutation. After 3 months of therapy, NGS no longer demonstrated the *IDH2* p.R172K mutation at significant levels in the BM sample obtained during clinical remission, whereas subsequent analysis by superRCA and ddPCR assays revealed significant levels of this mutation. All the following samples have been positive by all three detection techniques, consistent with the patient's clinical relapse. **C** Patient UPN 125 was followed by analyses of the *BCORL1* p.Q1039* mutation. The NGS and retrospective superRCA analysis revealed high levels of the *BCORL1* p.Q1039* mutation in the diagnostic samples collected on Day 0, but in samples collected during clinical remission from Day 97, the mutation levels had been suppressed to background by NGS and to $10^{-4}$ by superRCA. **D** Patient UPN 125 was followed with respect to the *DNMT3A* p.R882C mutation. Analysis by NGS and superRCA showed that the *DNMT3A* p.R882C mutation was initially present at a high frequency (MAF 0.38 and 0.44, respectively). and then persisted after treatment despite morphological remission as previously described for preleukemic mutations affecting *DNMT3A* mutations. The patient subsequently relapsed to an MDS phase without blast excess. **E** UPN126 was diagnosed with primary myelofibrosis two years prior to AML diagnosis. The bone marrow sample at AML diagnosis (day 0) revealed high levels of the *IDH2* p.R140Q mutation as recorded via NGS, superRCA and ddPCR. Retrospective analysis of the initial bone marrow sample at the myelofibrosis stage was negative for the *IDH2* p.R140Q mutation by superRCA and ddPCR. SCT was performed on day 139, at which point considerable levels of the mutation remained. The *IDH2* p.R140Q mutation was detected at low levels by superRCA approximately three months post SCT, whereas ddPCR failed to identify the detectable levels of the mutation at that timepoint. All subsequent samples have been negative in blood and bone marrow by superRCA and ddPCR, indicating a complete remission.

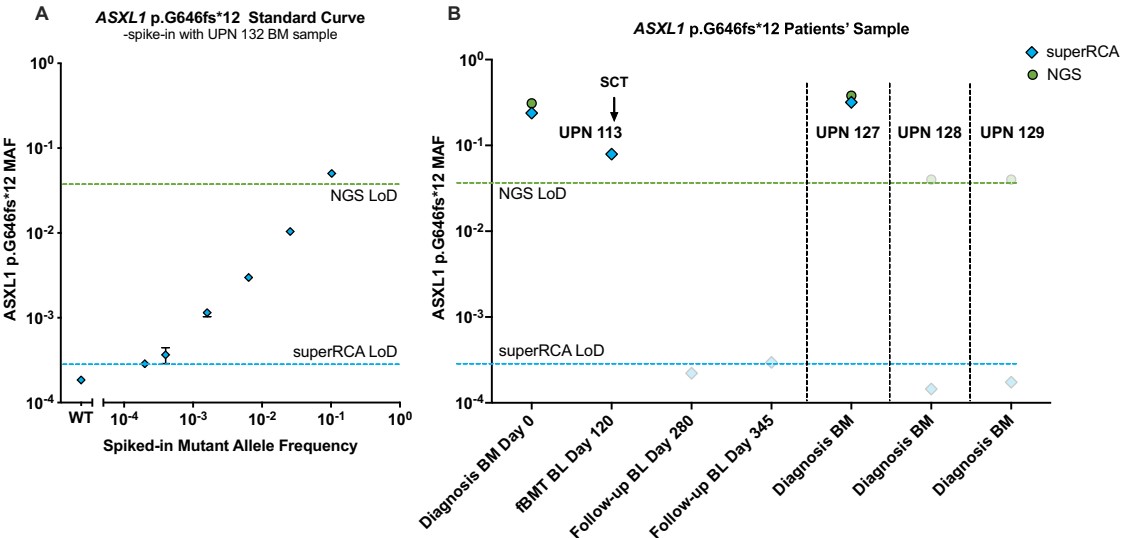

**Fig. 7 Analysis of the GC-rich *ASXL1* p.G646fs\*12 mutation in hematological malignancies. A** Standard curve of genomic DNA with the GC-rich *ASXL1* p.G646fs\*12 mutation, serially diluted 4-fold in wild-type gDNA and analyzed by superRCA in duplicates. **B** superRCA analysis of BM and blood samples from four individuals with AML and MDS, positive for the *ASXL1* p.G646fs\*12 mutation by NGS analysis.

samples; and with extreme accuracy in order to minimize risks of cross reactivity for far more abundant sequence variants. It is also imperative that authentic reaction products cannot be mistaken for any form of assay background. We present a solution to these challenges using a convenient molecular assay together with generally available instrumentation, and we demonstrate its application for monitoring leukemia patients at high precision and sensitivity.

Over coming years, tumor patients will increasingly have their malignancies sequenced and the mutations cataloged, thereby providing excellent tumor-specific markers to monitor the disease by analysis of e.g., plasma, cerebrospinal fluid, or as shown herein among cells in BM and among nucleated cells in peripheral blood. We demonstrate that a sufficiently precise and efficient method for detecting mutant gene sequences can guide therapy of tumor patients, permitting early detection potentially counteracting recurrences before tumor load increases, or offering a more confident diagnosis of stable remission. We show that in AML and the related disease MDS, peripheral blood may provide equally valid samples for monitoring patients as less accessible BM samples that are more unpleasant for the patient. Testing of peripheral blood can greatly simplify monitoring of patients, allowing sampling at shorter intervals to guide patient management.

Here, we compared the superRCA procedure to NGS and ddPCR assays currently in clinical use. NGS can be adapted to detect lower frequency mutations than shown herein, but at a greater cost and the turn-around time is long for NGS. Similarly, ddPCR is not limited to the 20,000 emulsion droplets aimed at here, but it still falls short in sensitivity compared to that resulting from the large number products generated in a typical of superRCA reaction (See Tables S1, S5 and Supporting material). In establishing the superRCA procedure we targeted a set of 12 mutations in the same DNA samples, including both point mutations and deletions, and we benchmarked superRCA against NGS and ddPCR. The superRCA technique is suitable to survey large sets of mutations that are known to be present in a patients malignant cells. Besides the initial PCR step used herein, large sets of target sequences can be captured in circular form in parallel using gap-fill padlock probes, also referred to as molecular inversions probes[37,38]. Padlock probes lend themselves for

application in very high multiplex, both for target capture and genotyping, since only intra-molecular probe reactions yield circular DNA products that can then be recorded via RCA, while ligation reactions between pairs of probes fail to yield amplifiable DNA circles.

We furthermore illustrate in Fig. 3C that the highly repeated sequence motifs in individual superRCA products render these suitable for combinatorial labeling with fluorophore-labeled detection probes for multiplex readout. In our illustration only two fluorophores were used but higher levels of multiplexing should be possible with the help of more fluorophores and labeling ratios, using standard flow cytometers.

superRCA assays build on the well-established ability of oligonucleotide ligation reactions in general and specifically padlock probes to discriminate among many target sequence variants, including single nucleotide variants, insertions and deletions and variable lengths of repeated sequences[28,37,39,40].

The low detection limit and high precision of superRCA are consequences of the highly selective genotyping of the repeated target sequences in combination with the large numbers of products that may be conveniently analyzed by flow cytometry. In combination, these properties present important clinical advantages. As illustrated by patient UPN125, NGS-analysis failed to detect the remaining *IDH2* p.R172K mutation after initial treatment which was therefore paused, although later superRCA and ddPCR analyses both clearly revealed the remaining malignant clone, subsequently leading to a relapse for this patient. In patient UPN126, the first ddPCR-analysis after SCT was reported negative for *IDH2* p.R140Q, although the mutation was still detectable at low levels by superRCA analysis. Even low levels of remaining leukemic markers in the post SCT-setting would prompt clinical action, mainly by reducing immunosuppressants to boost the immunological effect of the SCT in order to eradicate remaining malignant clones that risk giving rise to leukemic relapse.

Equally important is high accuracy in the genetic categorization of leukemia patients. The presence of an *ASXL1* mutation in the absence of a favorable-risk AML subtype places the patient in the adverse genetic risk category according to the ELN risk classification[7], thereby influencing treatment decisions. False negative detection of a high risk mutation such as *ASXL1*, as indicated in NGS-analysis for patients UPN128 and UPN129,

could therefore significantly influence patient management and outcome.

The superRCA assay procedure is suitable for routine use by the virtue of its high sensitivity and simplicity. The 3-hr protocol only requires a sequence of five additions to a DNA sample, separated by incubations, before reaction products are analyzed using a standard flow cytometer. Since completing this study we have automated the additions and incubation steps of the assay using a small lab robot. We demonstrate that each starting DNA circle results in a distinct, easily detected digital object of micrometer dimensions in an accurate process. Typically, more than 5 million such objects are generated in each reaction, allowing excellent quantitative precision with no need for specialized equipment besides a flow cytometer, counting products from each sample in a few minutes. Target sequences prone to mutation in tumors, such as GC-rich regions, may present challenges for sequence distinction, but we demonstrate that the superRCA procedure offers broad target compatibility.

As previously explained, the circumstance that padlock probe-based genotyping is performed on RCA products with hundreds of concatemeric copies of sequences to be interrogated means that occasional mistyping will not affect the overall genotyping of an individual RCA product, accounting for the very high sensitivity for rare mutant sequences. We demonstrate the ability to detect single-nucleotide mutations present at a ratio of 1:100,000 to the wild-type sequence. This corresponds to the correct genotyping of one in $10^{15}$ nucleotides in genomic DNA samples. Remaining risks for mistyping have to do with the risk for polymerase error during amplification of the target sequence, but the judicious choice of a high-fidelity polymerase maintains this risk below our reported detection threshold (see supplementary discussion).

The analysis of ultralow frequency mutations requires access to considerable amounts of DNA, which are easily obtained in bone marrow and whole blood samples. For liquid biopsy applications using plasma DNA, only small amounts of DNA are available. Here superRCA assays may instead present advantages in future applications by allowing simultaneous analysis of multiple mutant sequences in a patient sample (Fig. S8), with either joint or separate labeling for detection by flow as desired (See Fig. 3C), thereby enhancing detection sensitivity.

The superRCA mechanism demonstrated herein is also promising for many other applications where localized digital detection is required, including for in situ detection of rare mutant DNA or RNA sequences, and the technology may thus improve prospects for analyzing small amounts of mutant nucleic acids under a wide range of circumstances.

## Methods

**Ethical statement**. This study was approved by the Regional Ethics Committee of Uppsala-Örebro (2014/233, 2019/00130) and Stockholm (2017/2085-31/2).

**Extraction of genomic DNA**. DNA was extracted from BM cells or of whole blood using the QIAamp DNA Blood mini kit (Qiagen cat.51104) and eluted in 50 μL elution buffer.

**High fidelity PCR pre-amplification**. Sequences of interest in genomic DNA were amplified with SuperFi DNA polymerase (Thermo Scientific) in 50 μl PCR reactions containing 1X SuperFi buffer, 0.2 mM dNTP, 500 nM Fwd/Rev PCR primers, 330 ng gDNA and 0.04 U/μl SuperFi DNA polymerase. The PCR program was as follows: 98 °C for 30 s, 15 cycles of 98 °C for 15 s, 60 °C for 30 s, 72 °C for 10 s, and a final elongation at 72 °C for 5 min.

**Ligase-mediated circularization**. In total, 1 μl of amplified PCR products was diluted to 50 μl with MQ water, and 0.5 μl of this diluted PCR product was mixed with a 20 μl ligation solution containing 50 mM NaCl, 10 mM MgCl₂ and 10 mM Tris-Cl pH 8.0 (Rarity Bioscience), 10 nM of ligation template, complementary to both ends of one strand of the amplification products, 0.5 mM NAD (Sigma)

and 2 U Ampligase (Lucigen). The mixtures were incubated at 95 °C for 5 min, followed by 58 °C for 30 min.

**Target sequence amplification**. Circularized strands of PCR products containing target nucleotide positions were amplified by RCA. 5 μl of RCA buffer containing 50 mM NaCl, 10 mM MgCl₂, 10 mM Tris-HCl pH 8.0 (Rarity Bioscience), 1.8 mM dNTP (Invitrogen) and 2.5 U Phi29 polymerase (New England Biolabs) was added to the circularized products. The reactions were incubated at 37 °C for 30 min, then 65 °C for 10 min.

**Genotyping of RCA products**. Padlock probes were hybridized to first-generation RCA products and ligated in a sequence-specific manner, by adding 5 μL ligation mix containing 50 mM NaCl, 10 mM MgCl₂, 10 mM Tris-HCl pH 8.0 (Rarity Bioscience), 3 mM NAD (Sigma), 2.5 U Ampligase (Lucigen), and 60 nM genotyping padlock probe pairs to the reaction mixtures, incubating at 55 °C for 30 min.

**Secondary RCA**. Ligated padlock probes, encircling the first-generation RCA products, were amplified in a secondary RCA reaction. 30 μL RCA mixture containing 50 mM NaCl, 10 mM MgCl₂, 10 mM Tris-HCl pH 8.0 (Rarity Bioscience), and 6 U Phi29 DNA polymerase (New England Biolabs) was added to the genotyping ligation mixtures and incubated at 37 °C for 10 min. Then 5 μL of the reaction mixture containing 2.4 mM dNTP (Invitrogen), and 1.3 μM primers was added to the reaction mixtures, and the reactions were incubated at 37 °C for 30 min.

**Digital recording of superRCA products by flow cytometry**. The final reaction mixtures containing superRCA products were diluted into hybridization buffer containing 100 nM fluorophore-labeled oligonucleotide (Integrated DNA Technologies) probes specific for the different superRCA products, in 10 mM Tris-HCl pH 8.0, 10 mM MgCl₂ and 50 mM NaCl (Rarity Bioscience) to a final volume of 250 μL. The solutions were applied onto the CytoFlex flow cytometer (Beckman Coulter) and superRCA products were counted at "Medium" speed (30 μL/minute) for 150 s per sample.

**ddPCR IDH assay**. The QX200 AutoDG droplet digital PCR system (Bio-Rad) was used for ddPCR assay analysis. Primers/Probes for IDH variants were designed using the Bio-Rad online tool for ddPCR. Consumables and reagents for ddPCR were purchased from Bio-Rad and used according to the manufacturer's instructions. In short, PCR reaction mix was prepared in 22 μL using 1x ddPCR Supermix for Probes (no dUTP), 900 nM primers, 250 nM each probe (FAM and HEX) and the amount of DNA stated in each experiment. The reaction were then partitioned into ca. 20,000 droplets using the AutoDG from Bio-Rad, followed by PCR with 55 °C annealing/extension temperature. The droplets were then read with the QX200 Droplet Reader (Bio-Rad). Analysis of ddPCR data was done with QuantaSoft Analysis Pro software (Bio-Rad).

**TruSight-myeloid panel from Illumina**. Library preparation was performed with 50 ng input DNA using the clinically validated Trusight Myeloid Sequencing Panel (Illumina, San Diego, California) according to the manufacturer's instructions. This panel consists of 568 unique amplicons (~250 base-pairs) covering either hotspots or full coding region of 54 genes that are frequently mutated in myeloid neoplasms. After purification each library was quantified using the Qubit dsDNA HS Assay kit and the fragment size distribution was assessed using an Agilent 2200 Tapestation system (Agilent Technologies) with the high sensitivity D1000 Screen Tape. Equimolar libraries (6 per run) were then pooled, denatured and sequenced on the Illumina Miseq platform using reagent kit v3 chemistry, as specified by the manufacturer.

**Sequencing data analysis**. Amplicon mapping to human genome reference GRCh37/hg19, soft clipping of primer sequences, alignment and variant calling was performed using the TruSeq Amplicon analysis as implemented in the MiSeq Reporter software v2.6. Reads were not stitched. The Illumina pipeline was complemented with pindel 0.2.5b8 to search for larger insertions, deletions and internal tandem duplications. Regions with low coverage (<100×) were reported using in house algorithms to identify any clinically important regions where the analysis failed to produce data. Variants present in more than 15% of 1287 clinical routine samples were flagged as potential artifacts and were only included if the detected variant allele frequency was significantly higher than noise levels for that position. Common polymorphisms, intronic variants, variants with coverage below 500×, and variants below 5% VAF were filtered using Variant Analysis (QIAGEN). Remaining variants were annotated and assessed by hospital geneticists using QIAGEN Clinical Insight Interpret (QIAGEN). Sequence alignments around variants of interest were manually inspected using Integrative Genomics Viewer (Broad Institute, Cambridge, MA). All identified variants were further manually annotated using Alamut Visual software (version 2.11).

**Patient samples**. The cohort included 54 patients with myeloid neoplasias (Acute Myeloid Leukemia ((AML)) including secondary AML and 1 case of Acute Promyelocytic Leukemia (APL)), Myelodysplastic Syndrome (MDS) and myeloprolifeative neoplasia (MPN). The cohort comprised 23 females (43%) and 31 males (57%) and patients had a median age of 70,5 years (range 26–87). Participants were recruited solely based on prevalence of selected mutations. Mutational data was known from clinical diagnostic workup or prior mutational analysis, all the participants of this study were informed and consent was obtained from every participants. There is no financial compensation made to the participants of this study.

**Statistics and reproducibility**. The SEM data presented in Fig. 2A was repeated three times independently with similar results and The Flow data in Fig. 2C was repeated twice independently with similar results. No statistical method was used to predetermine sample size and No data were excluded from the analysis.

**Reporting Summary**. Further information on research design is available in the Nature Research Reporting Summary linked to this article.

## Data availability

The data supporting the results in this study are available within the paper, source data file and supplementary information. Source data are provided with this paper.

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

## Acknowledgements

The research was funded by The Swedish Research Council, the European Research Council under the European Union's Seventh Framework Programme (FP/2007-2013)/ERC Grant Agreement n. 294409 (ProteinSeq), IngaBritt och Arne Lundbergs Forskningsstiftelse, Vinnova Medtech4Healthmedtech4health-collaborative projects for improved health (2019-01464), The Swedish Foundation for Strategic Research (SB16-0046), Torsten Söderbergs Stiftelse (M130/16) and The Swedish Cancer Society (19 0384 Pj). We thank Tomas Edgren and Linus Bosaeus for valuable comments on the manuscript.

## Author contributions

L.C. conceived the superRCA idea, designed and performed the superRCA assays, analyzed the data and wrote the manuscript. A.E. helped with the sample collection, data analysis and written manuscript. S.W. and T.P. helped with the DNA sample preparation and performed the ddPCR and NGS analysis of the patient samples. S.L. provided with the patient samples and input for the paper. L.C. and U.L. conceived the superRCA idea, analyzed the data and wrote the manuscript.

## Funding

## Competing interests

L.C. and U.L. are inventors of a patent application (GB 2019074.0) regarding the superRCA technology and they are founders of the company Rarity Bioscience, pursuing this technology. The remaining authors declare no competing interests.
