## [Peer Review File · Nature Communications]

Reviewers' Comments:

Reviewer #1:

Remarks to the Author:

In this study Chen Lei and co-authors described the use of SafeLock assays for detection of specific DNA sequence variants at high sensitivity and specificity. The assay is based on targeted PCR amplification of the region of interest, conversion to DNA circles, subsequent rolling circle amplification and probing with Padlock probes specific for mutant or wild-type sequences that can be inspected via flow cytometry. Here, this assay was used in diluted cell lines for specific mutations, 47 samples with myeloid malignancies and in consecutive follow up samples from 3 AML cases and mutations in IDH1, IDH2 and ASXL1 were investigated. Although, the results successfully demonstrated a very high sensitivity and specificity, I have some major concerns related to the identification of multiple type of mutations in different genes and thus in its application for diagnostic purposes.

Major comments

- 1) This assay is based on previous knowledge of specific mutations and thus it limits the discovery of new mutations that can occur during clonal evolution under therapeutic pressure. It is not clear from the manuscript how many mutations can be simultaneously investigated in the same DNA sample or if viceversa for each mutation a distinct reaction is required. An exhaustive panel of recurrent mutations in myeloid malignancies should be assessed and evaluation of clonal heterogeneity should be evaluated and discussed.
- 2) The study is mainly focused on detection of missense mutations, with the exception of ASXL1p.G646fs*12 mutation. Which are the sensitivity and specificity of detection of other common frameshift mutations in myeloid malignancies? For example, mutations in TP53, DNMT3A, BCOR, etc.?
- 3) Have the authors used this assay for detection of gene fusions?

Reviewer #2:

Remarks to the Author:

In the manuscript by Chen et al, entitled "Ultra-sensitive monitoring of leukemia patients using SafeLock mutation detection assays", the authors describe an approach to detect SafeLock PCR products using flow cytometry. SafeLock PCR product can be detected with a mutant allele fraction as low as $10e-5$ (0.001%). The assay can multiplex 4 specific mutations and detect rare mutations in clinical samples similar to ddPCR. Overall, the manuscript is concise and easy to read, but lacks appropriate method details.

Major

1. The claim is made that SafeLock is more sensitive than standard ddPCR-based methods, however some details of the experimental comparisons are not clear. The sensitivity of ddPCR is a function of the total number of droplets analyzed. In most clinical labs multiple wells (reactions) are performed to maximize the number of droplets analyzed and therefore increase sensitivity. How many droplets were analyzed and how many PCR reactions were performed on each sample? How many additional reactions (droplets) are required to reach the same LOD described for SafeLock with ddPCR, or is it not possible to achieve the same LOD with ddPCR regardless of the number of droplets analyzed? For example, in Figure 4, different amounts of input DNA was used for SafeLock and ddPCR. Generating this figure using the same amount of DNA for each assay would be more meaningful.
2. Much emphasis is placed on the potential clinical utility of this method. Please compare the potential clinical benefits of the SafeLock MRD method to other more established molecular methods such as UMI-based deep sequencing including the newer TwinStrand approach. It would be useful to see a relative cost comparison of methods and a comparison of turn-around times in addition to differences in LOD.
3. Figure 4 and S4. It appears that SafeLock MAFs are consistently lower than NGS or ddPCR for MAFs $> 10e-2$ (i.e., 1%). Please provide a MAF correlation plot for SafeLock vs. ddPCR, and SafeLock vs. NGS using all the data with MAFs $> 10e-2$ from Figure 4 and S4 and using a linear MAF scale from 0-100% MAF on both axis instead of a log scale.
4. Figure S4B. Please comment on the reason why SafeLock did not detect mutations in patients 204 and 221.

5. How was the LOD calculated for each of the figures for NGS, ddPCR, and SafeLock?

Minor

1. Figure 1C. The middle panel contains some yellow products. The text on line 152 says there are no mixed-color products. Please address this discrepancy.
2. Figure 3C. Please explain why you get 7 different colors when 2 fluorophores are used. What do the different colored products represent? What do the different clusters of products within the same color represent?
3. Figure S3ii. What are the black colored events, are they products containing both WT and Mut sequence?
4. Figure 4. Please provide representative images from the flow cytometry plots and ddPCR plots in the supplement for variants with MAFs at 10^{-4} and 10^{-5} .
5. On line 208 and 209, please also provide the number of diploid cells that correspond to the haploid genome numbers.
6. SafeLock multiplexed four IDH mutations. What is the maximum number of products that could be multiplexed with SafeLock?
7. Line 252 'venetoclax' spelling and italicize all gene symbols throughout.
8. Line 253. Only a fraction of AMLs express a CD34+, CD117+, CD33+, CD13 heterogen, HLA-DR+ and CD56+ phenotype. Was testing limited to only patients with this phenotype?
9. Line 269 I would not use the term 'MDS' in this situation as the diagnosis requires specific criteria that are not demonstrated. IDH1/2 mutations are also generally rare in MDS. 'Clonal hematopoiesis' would be a more appropriate term.
10. Have the authors tried amplifying CEBPA, a GC rich gene that is difficult to sequence.
11. Can SafeLock be used to detect indels greater than 1 nucleotide in size?
12. Methods. Please provide the amplicon size for the 4 IDH mutations and the ASXL1 mutation. Please add a supplemental table listing all the primer sequences used (pre-amplification, first RCA, and secondary RCA reactions), the sequence of the ligation template, the sequence of the Padlock probes, sequence for the fluorophore labeled oligonucleotide probes specific for the SafeLock products, and explain how were the fluorophores conjugated to the probes.
13. Methods. Line 602, is it correct that 0.05 microliters of PCR product was used.
14. References – Consider adding duplex sequencing, PMID: 22853953, and rare mutant cell detection, PMID: 30207916. References 14 and 23 are duplicates.

Reviewer #3:

Remarks to the Author:

The manuscript by Chen et al. described a Safelock assay for the ultra-sensitive detection of rare single-nucleotide mutations. The strategy was established based on their previous works that padlock probes could be used to discriminate SNVs and amplify detection signals in situ. Compared to their previous works, a PCR preamplification and two RCA steps were included in the same assay to enhance the sensitivity, and flow cytometry was used for the detection of final RCA products. A remarkable 1:100,000 sensitivity was achieved for rare mutations in AML, which was significantly better than the state-of-the-art techniques, such as ddPCR and NGS. The assay was finally validated against bone marrow aspirates and blood samples from 47 AML patients. Given the remarkable clinical performance, I believe the Safelock assay is of potential interest to the readers of Nature Communications. My detailed comments were listed as follows:

1. I think it will be beneficial to include a brief history of the development of padlock probes and RCA in the introduction with citations of milestones (e.g., Nucleic Acids Res., 2003, 31, e103; Nat. Methods, 2004, 1, 227; Nucleic Acids Res., 2005, 33, 8e70) for readers who are not familiar this panel of technologies. When deployed for genotyping, the authors shall also discuss how much improvement when using the Safelock compared to conventional padlock probes.
2. What is the absolute limit of the detection (LOD) of Safelock for quantifying nucleic acid targets? The sensitivity in the manuscript referred to the frequency of mutated sequence in the high concentrations of wild-type. It remains unclear how many copies of mutated sequences could be detected in the sample?
3. A related question to comment-2 was the comparison between Safelock and ddPCR. Why the different amounts of starting genomic DNA was used (330 ng for Safelock and 66 ng for ddPCR)?

Is it because the LOD of Safelock was worse than ddPCR?

4. The authors shall also characterize the ligation efficiency for circulating PCR amplicons. This could be a limiting step for the assay. How did the authors avoid the loss of a minute amount of rare mutation through either failed ligation or the formation of dimers? PAGE analysis shall be performed and included to show the quality of the ligation product.

5. I don't think the Safelock can be classified as a "simple" assay, as it includes nucleic acid extraction, PCR pre-amplification, ligation to form the circular probes, the first round of RCA and the second round of RCA, labeling of hybridization probes, and a final flow cytometry analysis. This is considerably more complicated than RT-PCR or ddPCR and would be very difficult to be fully automated and concealed to avoid cross-contamination. The authors shall discuss this and other disadvantages and potential solutions for their assay.

6. As an ultrasensitive assay for rare mutations, it is also critical to evaluate the clinical specificity to avoid false-negative tests. Ideally, comparison with deep sequencing shall be performed to cross-validate the results in Figure 4b and clinical tests for AML samples.

7. Figure 4a-C, the "mutant allele frequency=0" data of ddPCR was missing.

We are grateful for the constructive advice for our paper by the three reviewers, and we feel the paper has been further strengthened by the amendments we have made in response to these comments, as described in our responses to the comment below.

We also wish to point out that we have decided to refer to the new technique described in our paper as superRCA (referring to the two generations of RCA for each product) rather than SafeLock as in the previously submitted version of the manuscript.

Reviewer #1 (Remarks to the Author):

In this study Chen Lei and co-authors described the use of SafeLock assays for detection of specific DNA sequence variants at high sensitivity and specificity. The assay is based on targeted PCR amplification of the region of interest, conversion to DNA circles, subsequent rolling circle amplification and probing with Padlock probes specific for mutant or wild-type sequences that can be inspected via flow cytometry. Here, this assay was used in diluted cell lines for specific mutations, 47 samples with myeloid malignancies and in consecutive follow up samples from 3 AML cases and mutations in IDH1, IDH2 and ASXL1 were investigated. Although, the results successfully demonstrated a very high sensitivity and specificity, I have some major concerns related to the identification of multiple type of mutations in different genes and thus in its application for diagnostic purposes.

Major comments

1) This assay is based on previous knowledge of specific mutations and thus it limits the discovery of new mutations that can occur during clonal evolution under therapeutic pressure. It is not clear from the manuscript how many mutations can be simultaneously investigated in the same DNA sample or if viceversa for each mutation a distinct reaction is required. An exhaustive panel of recurrent mutations in myeloid malignancies should be assessed and evaluation of clonal heterogeneity should be evaluated and discussed.

It is correct that the assay we describe is not suited for detecting previously unknown mutations, and we point out on line 443-446 that the intended application is to sensitively follow the course of malignant disease by monitoring minimal or measurable residual detectable (MRD) in the increasingly common situation where the mutational status of a patient's malignancy is already known, at least in part. We believe this assay can meet a growing clinical need to sensitively monitor tumor patients and their responses to therapy, as we illustrate by monitoring leukemia patients under therapy.

Regarding the question how many mutations can be investigated simultaneously, we now mention the fact that the padlock probe ligation step lends itself for application in high multiplex (line 446 and via references 38 and 39) both for target capture and genotyping. This is so because only intra-molecular probe reactions yield circular DNA products that can then be amplified via RCA, while ligation reactions between pairs of probes fail to yield amplifiable DNA circles. Therefore, risks of cross reactions do not grow rapidly with the use of more padlock probes. Furthermore, we illustrate in Figure 3C that the many repeated sequence motifs in individual superRCA products means that the number of products that can be distinguished by flow cytometry is not limited by the number of distinguishable fluorophores, but that combinations of fluorophores can be used to reach higher levels of multiplexing at the readout stage. In our illustration only two fluorophores were used, but higher levels of multiplexing are possible using standard flow cytometry with more fluorophores and more labeling ratios.

With respect to the selection of mutations for our study, the IDH panel for AML was designed in consultation with our clinical colleagues with the aim to monitor tumor burden. There was no intention in this study to monitor clonal heterogeneity by using an exhaustive panel of mutations. Regarding the ability to analyze diverse mutations using the described procedure, please see response to question 2 below.

When it comes to the numbers of mutations analyzed in parallel in our paper, we now explain more clearly on line 216 that in the experiments underlying figure 4, different PCR amplicons were generated and circularized, corresponding to the target regions for the four mutations. For the following padlock probe-based genotyping step, the circularized reaction products were divided in separate reactions for each investigated mutation, followed by separate readout via flow cytometry.

2) The study is mainly focused on detection of missense mutations, with the exception of ASXL1p.G646fs*12 mutation. Which are the sensitivity and specificity of detection of other common frameshift mutations in myeloid malignancies? For example, mutations in TP53, DNMT3A, BCOR, etc.?

As to the question which classes of mutations can be detected using our method, we have added a paragraph discussing the generality of the approach with appropriate references on line 476. Briefly, the superRCA assay builds on the well-established ability of oligonucleotide ligation reactions in general and specifically padlock probe reactions to discriminate against mismatched target sequences. In the case of superRCA, the distinction is greatly enhanced by the fact that the task is to genotype hundreds of copies of the target sequence present in individual RCA products, ensuring essentially error-free genotyping by what we term a majority vote-mechanism. As we explain on line 124, the many tandem copies of the target sequence safeguard against the occasional mistyping reaction since this will be undetectable against a vast majority of correctly genotyped copies in each RCA product.

To illustrate the excellent ability to distinguish closely similar target sequences we include in the paper data for typing the ASXL1 p.G646fs*12 insertion (Fig. 7). This mutation represents an extreme case that presents great difficulty for conventional NGS/ddPCR assays, resulting in poor sensitivity for ASXL1 mutations. This is so because accurate typing requires the ability to discriminate between sequence variants with either 8 or 9 consecutive G residues, located in a very high-GC region (Figure 6A). For classical frameshift mutations where the sequences at the breakpoints are known, we can achieve an even better distinction than for missense mutations, since these sequence variants are more radically different, and therefore easier to distinguish compared to the closely similar single nucleotide variants that we illustrate in our other targeted mutations.

3) Have the authors used this assay for detection of gene fusions?

None of the mutations we have explored so far with the superRCA technique are gene fusions. However, provided that the sequence at the junction is identified, it is well known from prior padlock probe work that the distinction of such sequences from wildtype is a simpler task than the single-nucleotide exchange or length distinctions demonstrated in our paper.

Reviewer #2 (Remarks to the Author):

In the manuscript by Chen et al, entitled "Ultra-sensitive monitoring of leukemia patients using SafeLock mutation detection assays", the authors describe an approach to detect SafeLock PCR products using flow cytometry. SafeLock PCR product can be detected with a mutant allele fraction as low as $10e-5$ (0.001%). The assay can multiplex 4 specific mutations and detect rare mutations in clinical samples similar to ddPCR. Overall, the manuscript is concise and easy to read, but lacks appropriate method details.

Major

1. The claim is made that SafeLock is more sensitive than standard ddPCR-based methods, however some details of the experimental comparisons are not clear. The sensitivity of ddPCR is a function of the total number of droplets analyzed. In most clinical labs multiple wells (reactions) are performed to maximize the number of droplets analyzed and therefore increase sensitivity.

How many droplets were analyzed and how many PCR reactions were performed on each sample? How many additional reactions (droplets) are required to reach the same LOD described for SafeLock with ddPCR, or is it not possible to achieve the same LOD with ddPCR regardless of the number of droplets analyzed? For example, in Figure 4, different amounts of input DNA was used for SafeLock and ddPCR. Generating this figure using the same amount of DNA for each assay would be more meaningful.

As pointed out by the reviewer, the sensitivity of ddPCR is a function of the number of droplets analyzed, but also the risk of an incorrect genotyping result in individual droplets. We agree that the comparison with ddPCR may require further explanation, and we have added this information in line 225-230, line 243-250 and also in the Materials and Methods section, line 877. Briefly, in the ddPCR benchmark study, we used 132 ng total genomic DNA as input for a duplicate ddPCR run (see Figure 4 legend line 245). We are following the recommendations by the manufacturer Bio-Rad by using an input of 66 ng gDNA for each of the two ddPCR replicates to create 40,000 emulsion droplets that accommodate some 40,000 copies of haploid genomes. In our analysis of the ddPCR data, we observed a background of the ddPCR assay at a mutant allele frequency around 0.1% in a single ddPCR reaction. Pooling two ddPCR reactions (40,000 droplets) or five ddPCR reactions (100,000 droplets) cannot avoid this background since both ddPCR reactions would have the same background of 0.1% MAF for pure wild-type samples. Providing more DNA samples with multiple ddPCR reactions therefore cannot further improve mutation detection sensitivity.

We investigated the false positive rate by ddPCR. As shown in Figure S5 it was not possible to significantly decrease this rate, either by providing more DNA per well or by pooling data from up to 10 wells with the same amount of gDNA per well (Fig. S5).

Another factor, also discussed in the line 425-430 and Supplementary Material section line 778, is that the precision with which a mutation frequency can be estimated depends on the number of digital measurements, and the Poisson noise that results from chance variation of small numbers of measurements. ddPCR measures a few tens of thousands events, while with superRCA we typically count around a million events in around one and a half minute using a flow cytometer. As can be seen in Figure 4 in our paper, ddPCR measurements exhibits greater variability (CV) compared to superRCA due to Poisson noise, which reflects the number of counted objects. This improved precision of superRCA also has effects on the limits of detection since the Limit of Detection (LoD) is set at three CVs ($LoD = Mean(wildtype) + 3 \times SD(wildtype)$) over the average background and the improved precision of superRCA therefore admits a considerably lower detection threshold.

2. Much emphasis is placed on the potential clinical utility of this method. Please compare the potential clinical benefits of the SafeLock MRD method to other more established molecular methods such as UMI-based deep sequencing including the newer TwinStrand approach. It would be useful to see a relative cost comparison of methods and a comparison of turn-around times in addition to differences in LOD.

This is an important point and we have now introduced a detailed comparison between superRCA and other methods in Supplementary Material, line 772. The superRCA assay requires only 3.5 hr to establish the mutation frequency information starting from the purified DNA samples. The NGS workflow with TwinStrand has a typical one week turn-around time, at a considerably higher cost than for superRCA, while ddPCR cannot offer the sensitivity of superRCA, as we now detail in the Supplementary Material line 778.

3. Figure 4 and S4. It appears that SafeLock MAFs are consistently lower than NGS or ddPCR for MAFs > 10e-2 (i.e., 1%). Please provide a MAF correlation plot for SafeLock vs. ddPCR, and SafeLock vs. NGS using all the data with MAFs > 10e-2 from Figure 4 and S4 and using a linear MAF scale from 0-100% MAF on both axes instead of a log scale.

In ddPCR assays, the double positive emulsions (emulsion landing in top right corner) are counted as mutant events, while in the superRCA assay, we previously discarded those double positive events, resulting in an under-representation of the mutant population. We have now added results from a re-run using the same batch of pre-amplified libraries from these diagnostic samples where we also counted these double positive events and use the data to compare our results to those of ddPCR and NGS, demonstrating very good agreement between the three methods. This is now discussed in line 262 in the paper.

4. Figure S4B. Please comment on the reason why SafeLock did not detect mutations in patients 204 and 221.

We now discuss on line 282 that Patient 204 carries the IDH1 R132C mutation and Patient 221 is positive for the IDH2 R172K mutation according to the NGS analysis. Patient 204 was also shown to be positive for IDH1 R132C mutation by superRCA analysis (now it is Figure S8, panel A), and patient 221 was shown to be positive for IDH2 R172K by superRCA (Figure S8 panel D). Patients 204 and 221 also appear to be positive for IDH1 R132H by ddPCR in Figure S8 panel B, but these are likely to be false positive signals from the ddPCR IDH1 p.R132H analysis as the patient 204 (0.14%) and 221 (0.12%) only presents relatively low allele frequency of IDH1 p.R132H which is close to the LoD of ddPCR assay, while superRCA/SafeLock reported negative in the IDH1 p.R132H assay for these two patients.

5. How was the LOD calculated for each of the figures for NGS, ddPCR, and SafeLock?

We explain on line 251 that the LOD calculation was measured as $LoD = Mean(Wildtype) + 3 \times SD(wildtype)$, and we clarify that the LOD for ddPCR and superRCA were determined with the spike-in samples illustrated in Figure 4a.

Minor

1. Figure 1C. The middle panel contains some yellow products. The text on line 152 says there are no mixed-color products. Please address this discrepancy.

We now explain on line 164 that when pure wild-type or mutant samples were investigated, no mixed-color products were observed, demonstrating the unambiguous typing of the products. We point out that the yellow products seen in the 100:1 condition in Figure 1C are due to the random colocalization of the wild-type and mutant amplification products deposited on the glass surface.

2. Figure 3C. Please explain why you get 7 different colors when 2 fluorophores are used. What do the different colored products represent? What do the different clusters of products within the same color represent?

We now describe the labeling of superRCA products more clearly in line 206. Briefly, in Figure 3C, we used detection oligonucleotides labeled with the two fluorophores AL488 and AL647 at the indicated ratios to label the superRCA/SafeLock products. After labeling, the labeled superRCA products were pooled and analyzed by flow cytometry using standard instrument software. We used the software to display the 7 populations in distinct pseudo-colors.

3. Figure S3ii. What are the black colored events, are they products containing both WT and Mut sequence?

We agree this was unclear and we now explain on line 732 that the black events in Figure S9 are ones that do not fall into the gates, either P2 or P3 as indicated in Figure S3. We also point out that these events represent 1.59% of the total recorded superRCA products. In figure S9ii,

events displayed at the top right corner are considered double positives. For this particular high-MAF (5.6%) sample, we recorded approximately 853,000 events during a 150 s acquisition time (the detailed acquisition conditions are described in the Materials and Methods section, line 850). At a rate of 5700 events recorded per second, we expect around 1.59% double positives $(852776-791149-48043)/852776=1.59\%$. This corresponds nicely to the numbers of double positives observed at the top right of Figure S9ii). We also point out that the size of this double positive population can be decreased by using a slower acquisition speed or using a more dilute population of superRCA products to decrease the chance that both wild-type and mutant products would pass through the detector simultaneously. The effect of resolving the black population by diluting the solution with superRCA products is illustrated in the added Figure S10.

4. Figure 4. Please provide representative images from the flow cytometry plots and ddPCR plots in the supplement for variants with MAFs at $10e-4$ and $10e-5$.

As suggested, we now include the raw data images from the IDH2 p.R172K superRCA and ddPCR performance benchmark below in the Supplement Fig. S6A-D to demonstrate the feasibility of detecting even low frequency events by superRCA compared to a negative sample. The ratio between objects in gate P2 versus P2+P3 combined is proportional to the spike-in levels. Fig. S6E-H presents the corresponding ddPCR raw data for the IDH2 p.R172K mutation and a negative control.

5. On line 208 and 209, please also provide the number of diploid cells that correspond to the haploid genome numbers.

In line 228, we now mention that 330ng DNA corresponds to 100,000 human haploid genomes" or 50,000 diploid cells, and in line 216 we write that "66 ng DNA corresponds to 20,000 human haploid genomes or 10,000 diploid cells. Since the cells are cycling some will have replicated part or all of their genomes, so that the true number of cells for 330 ng is somewhat lower than 50,000.

6. SafeLock multiplexed four IDH mutations. What is the maximum number of products that could be multiplexed with SafeLock?

Please see responses to comment 1 by reviewer #1 and references to the changes of the manuscript therein. Briefly, in this AML study we demonstrated detection of 4 different mutations in one portion DNA sample, but the potential for multiplexing is considerably greater.

7. Line 252 'venetoclax' spelling and italicize all gene symbols throughout.

We have now corrected the spelling for 'venetoclax' in the main text, the gene symbols styles were following the Nature Communication style with italics.

8. Line 253. Only a fraction of AMLs express a CD34+, CD117+, CD33+, CD13 heterogen, HLA-DR+ and CD56+ phenotype. Was testing limited to only patients with this phenotype?

The patients did not have identical phenotypes, and we did not limit our testing to the phenotype above, although all three patients that were followed over time were positive for the common AML markers CD34 and CD117. We now describe their phenotypes in the supplementary (line 807) as follows:

UPN 124: CD34+, CD117+, CD33+, CD13 heterogenous HLA-DR+ and a subpopulation CD56+

UPN 125: CD34+, CD117+, CD33 dim, CD13 heterogenous MPO+, morphologic signs of dysplasia

UPN 126: CD34+, CD117+, CD13+, HLA-DR+

9. Line 269 I would not use the term 'MDS' in this situation as the diagnosis requires specific criteria that are not demonstrated. IDH1/2 mutations are also generally rare in MDS. 'Clonal hematopoiesis' would be a more appropriate term.

We agree that clonal hematopoiesis in the setting of peripheral cytopenia is a more accurate description, and we have made the change on line 326. IDH mutations are assumed to represent early driver events in the development of myeloid malignancies and have been reported in about 7% of MDS cases, with IDH2 as the more frequent type. The patient had a 3.5 year history of macrocytic anemia and leukopenia, progressing over time prior to AML, although some morphological signs of dysplasia were reported already at the initial presentation.

10. Have the authors tried amplifying CEBPA, a GC rich gene that is difficult to sequence.

We have not investigated CEBPA yet, but we aim to expand our assay portfolio by investigating more mutations present in high-GC regions. As we write in response to question 2 by Referee 1, we include as figure 7 one example of successful typing of a very challenging GC-rich sequence to illustrate this property of the superRCA technique.

11. Can SafeLock be used to detect indels greater than 1 nucleotide in size?

Please see our responses to this question in the context of question 2 by Reviewer #1. Briefly, indels of more than 1 nucleotide are easier to distinguish than the single nucleotide variants and the 1 nucleotide indel we demonstrate in the paper, since the ligation of padlock probes mismatched to such length variants are more efficiently inhibited.

12. Methods. Please provide the amplicon size for the 4 IDH mutations and the ASXL1 mutation. Please add a supplemental table listing all the primer sequences used (pre-amplification, first RCA, and secondary RCA reactions), the sequence of the ligation template, the sequence of the Padlock probes, sequence for the fluorophore labeled oligonucleotide probes specific for the SafeLock products, and explain how were the fluorophores conjugated to the probes.

The requested sequence information has now been added as Supplementary Table S2 and we describe in line 897 that oligonucleotides were ordered by a supplier with the fluorophores added.

13. Methods. Line 602, is it correct that 0.05 microliters of PCR product was used.

Yes, we use a limited amount of products from the pre-amplification step which allows us to detect many mutations in parallel reactions. We now describe this on line 823. In practice, the PCR products were diluted 10-fold in MQ water before being combined with the 1st ligation mix.

14. References – Consider adding duplex sequencing, PMID: 22853953, and rare mutant cell detection, PMID: 30207916. References 14 and 23 are duplicates.

Thanks for pointing this out – the duplicated reference has now been removed and we have added the suggested references in lines 92 and 388 respectively. Regarding sequencing

approaches for rare mutation detection, we have expanded this discussion, starting on line 89 and in the Supplementary Material line 778. We have also added the following references to bring the paper up to date: Michael etc. Proc Natl Acad Sci U S A. 2012 as reference 25 in line 92 and Eric J Duncavage etc, N Eng J Med 2018. is now cited as reference 35 in line 388.

Reviewer #3 (Remarks to the Author):

The manuscript by Chen et al. described a Safelock assay for the ultra-sensitive detection of rare single-nucleotide mutations. The strategy was established based on their previous works that padlock probes could be used to discriminate SNVs and amplify detection signals in situ. Compared to their previous works, a PCR preamplification and two RCA steps were included in the same assay to enhance the sensitivity, and flow cytometry was used for the detection of final RCA products. A remarkable 1:100,000 sensitivity was achieved for rare mutations in AML, which was significantly better than the state-of-the-art techniques, such as ddPCR and NGS. The assay was finally validated against bone marrow aspirates and blood samples from 47 AML patients. Given the remarkable clinical performance, I believe the Safelock assay is of potential interest to the readers of Nature Communications. My detailed comments were listed as follows:

1. I think it will be beneficial to include a brief history of the development of padlock probes and RCA in the introduction with citations of milestones (e.g., *Nucleic Acids Res.*, 2003, 31, e103; *Nat. Methods*, 2004, 1, 227; *Nucleic Acids Res.*, 2005, 33, 8e70) for readers who are not familiar this panel of technologies. When deployed for genotyping, the authors shall also discuss how much improvement when using the Safelock compared to conventional padlock probes.

As pointed out by the reviewer there is a large body of literature on ligase-mediated genotyping and specifically on the use of padlock probes that provides a relevant background to the present paper. As recommended, we have now expanded this background on line 104 and onwards and by adding references Luo.J. *Nucleic Acids Res* 1996, Naner, J, *Nucleic Acids Res.*2003 and Larsson, C, *Nat. Methods* 2004. This prior art is cited in line 107, and the comparison to previous ligase-mediated genotyping strategies is briefly discussed on line 104and onwards.

2. What is the absolute limit of the detection (LOD) of Safelock for quantifying nucleic acid targets? The sensitivity in the manuscript referred to the frequency of mutated sequence in the high concentrations of wild-type. It remains unclear how many copies of mutated sequences could be detected in the sample?

We have expanded the discussion about the probability of detecting mutant sequences on line 494 and onwards. Briefly, the absolute limits of detection of mutant molecules were determined by spike-in experiments. The limited cycles of pre-amplification by PCR serves to expand the number of copies from the initial genome sequence input, and thus increase the chance to detect mutant DNA molecules in the sample. For the spike-in experiments shown in Figure 4, we used a total DNA input of 660 ng which corresponds to 200,000 haploid genomes at 1 in 100,000 levels to ensure we can detect the mutant molecules reliably. In the flow data we typically record more than 10 mutant events at the 1/100,000 spike-in level (Supplementary Figure S6A-D).

3. A related question to comment-2 was the comparison between Safelock and ddPCR. Why the different amounts of starting genomic DNA was used (330 ng for Safelock and 66 ng for ddPCR)? Is it because the LOD of Safelock was worse than ddPCR?

This subject has been described more thoroughly on line 225 and also in the Supplement Figure S4 and S5, In Figure S4, from the manufacturer's IDH assay validation data, in the condition that 132 ng (equivalent to 40,000 human haploid copies) of human genomic DNA samples per reaction were used in the spike-in validation, the ddPCR assay were still not able to push the sensitivity below 1/10,000(Bio-Rad ddPCR IDH2 p.R172K assay validation report

<https://www.bio-rad.com/digital-assays/assay-detail/dHsaCP2000060>). The superRCA assay is able to use 330 ng per reaction and deliver a sensitivity of 1 in 100,000. The ddPCR IDH2 p.R172K assay (Fig.S4D) had a LoD (dashed line) of around 0.04% with a total DNA input of 40,000 copies, while on the assumption that the mutation detection sensitivity can be improved by providing more DNA samples, the 40,000 genomes could theoretically reach to 0.0025% MAF for the IDH2 p.R172K assays, which is 10 times lower than the assay validation data from Bio-Rad.

We have now performed a false positive rate investigation on ddPCR platform. The data in Figure S5 demonstrates that it is difficult to push the sensitivity on ddPCR platform by providing more DNA content per well or pooling data from up to 10 wells with 33ng gDNA per well. (Fig. S5).

IDH2, WT for p.R172K, *Human*

Fractional Abundance (%)

Dilution series of mutant DNA in ~40,000 copies of wild type DNA background.

Figure S4D: IDH2 p.R172K assay validation from Bio-Rad with 40,000 copies of wild-type DNA input.

4. The authors shall also characterize the ligation efficiency for circulating PCR amplicons. This could be a limiting step for the assay. How did the authors avoid the loss of a minute amount of rare mutation through either failed ligation or the formation of dimers? PAGE analysis shall be performed and included to show the quality of the ligation product.

We now clarify this important point in Fig S1 and its legend. Because of the initial low cycle PCR there is no need for 100% efficient conversion to DNA circles, but circularization is highly efficient with negligible production of dimer strands.

5. I don't think the Safelock can be classified as a "simple" assay, as it includes nucleic acid extraction, PCR pre-amplification, ligation to form the circular probes, the first round of RCA and the second round of RCA, labeling of hybridization probes, and a final flow cytometry analysis. This is considerably more complicated than RT-PCR or ddPCR and would be very difficult to be fully automated and concealed to avoid cross-contamination. The authors shall discuss this and other disadvantages and potential solutions for their assay.

We agree that RT-PCR is a simpler assay, but with substantially lower performance for the task at hand. As we point out on line 85-87, downstream of the DNA isolation and initial PCR, all reactions for the superRCA procedure are initiated by reagent additions to the same reaction

tube before the final flow cytometric readout and with no need for specialized lab equipment. In fact, after completing this manuscript we have automated this procedure in a microtiter format using a standard lab robot. We now analyze 96 different samples with minimal hands-on time. Regarding the contamination issue, standard measures such as performing PCR with the addition of UNG and dUTP can be used to prevent carry-over from previous runs (Longo MC, Berninger MS, Hartley JL. Use of uracil DNA glycosylase to control carry-over contamination in polymerase chain reactions. *Gene* 1990;93:125-128.)

6. As an ultrasensitive assay for rare mutations, it is also critical to evaluate the clinical specificity to avoid false-negative tests. Ideally, comparison with deep sequencing shall be performed to cross-validate the results in Figure 4b and clinical tests for AML samples.

The patient samples analyzed in Figure 5 were diagnostic obtained from patients diagnosed with AML, and the samples had all been examined through clinical NGS analysis when levels of mutant cells in bone marrow were high. We present that data in Figure 5 as green bars. The superRCA data clearly indicate that we correctly identified the mutation with no false negative or positive observations. Regarding the comparison to deep sequencing, we believe this comparison is best made by reference to state-of-the-art publications from leading labs, and we now discuss this comparison at some length in Supplementary data, line 778.

7. Figure 4a-C, the “mutant allele frequency=0” data of ddPCR was missing.

This mistake has now been corrected.

Reviewers' Comments:

Reviewer #1:

Remarks to the Author:

The authors addressed all my comments and I do not have further concerns.

Reviewer #2:

Remarks to the Author:

Thank you for addressing my comments. I have no additional comments.

Reviewer #3:

Remarks to the Author:

I am generally satisfied with the response and revisions to my previous comments and concerns. I believe the manuscript is ready to be accepted in its present form.